# Assessment of Raisins Byproducts for Environmentally Sustainable Use and Value Addition

Mahmoud Okasha [1,*], Rashad Hegazy [2] and Reham Kamel [1]

1   Agricultural Engineering Research Institute (AEnRI), Agricultural Research Center (ARC), Giza 12611, Egypt; rehamkamel8541@gmail.com
2   Agricultural Engineering Department, Faculty of Agriculture, Kafrelsheikh University, Kafrelsheikh 33516, Egypt; rashad.hegazy@agr.kfs.edu.eg
*   Correspondence: mahmoudokasha1988@yahoo.com; Tel.: +0020-10-0313-3841

**Abstract:** This study investigated the potential and sustainable use of the biomass derived from various stages of the grape drying process. A total of eleven byproducts, each containing varying organic materials, were produced and subjected to testing. Ultimate analysis, as well as analyses of heating values, chemical composition, lignocellulose composition, total solids concentration and biogas production were performed with the recommended criteria and assessment methods. The results reveal that carbon (C), nitrogen (N), hydrogen (H), and oxygen (O) levels were significantly different among the byproducts. The ash content of byproducts 5–11 ranged from 3.56 to 5.11%, which was lower than the estimated values in the other byproducts. The analysis of higher heating value showed significantly higher calorific values for byproducts 10 and 11 ($22.73 \pm 0.08$ and $22.80 \pm 0.07$ MJ kg$^{-1}$, respectively). Byproducts 1–9 had lower sugar content than byproducts 10 and 11 (rejected raisins). Byproducts 5–9 had the lowest lignin content, and there were no significant differences in neutral detergent fiber (NDF) contents between byproducts 1–6. The highest accumulated biogas volume after 40 days was 11.50 NL L$^{-1}$ of substrate for byproduct group C (byproducts 10 to 11), followed by 11.20 NL L$^{-1}$ of substrate for byproduct group B (byproducts 5–9) and 9.51 NL L$^{-1}$ of substrate for byproduct group A (byproducts 1–4). It is concluded that byproducts consisting of biomass derived at different stages of raisin production may be an effective solid fuel and energy source. The amounts of volatile solids in the tested raisin processing byproducts indicated their appropriateness for pyrolysis conversion to a liquid product with high volatile content.

**Keywords:** raisins byproducts; ultimate analysis; heating values; biogas production

## 1. Introduction

Over 77 million tons (Mt) of grapes were produced in 2018 [1], and 7% of grape production is treated to produce dried grapes, yielding an annual production of 1.21 Mt of dried raisins for 2018/19 [2]. A great number of products derived from grape processing are available in the market, such as wine, juice, jam, and raisins [3], and to produce these products, a very large amount of waste is generated, which leads to negative environmental impacts when it is disposed [4]. Residues produced from juice or wine grape processing, such as grape pomace, skins, and seed, have been well studied for potential use as byproducts in different industries [5]. However, in processing grapes to produce raisins, only a few researchers have studied the various wastes generated during the different drying stages for their potential use as valuable byproducts. Many tons of waste result from drying grapes to produce raisins. There is much potential benefit in utilizing this waste to reduce the environmental impacts resulting from its mismanagement, including impacts on the environmental, economic, and social sectors [6].

Bioenergy production from bio-residues, energy crops, and other forms of biomass is a well-established technology, and is gaining more consideration among the most concerned stakeholders [7]. Wet oxidation could produce pulps with better enzymatic digestibility

from grape stalks and low-cost tannin-rich feedstock [8]. Grapes stalks and rachises represent 12% of the total winery residues and contain high cation levels. Therefore, conditioning treatments, such as biodegradation activators combined with water and high temperatures, are required before the stalks are used in biochar or bioenergy processes [9]. Dinuccio et al. [10] attributed the specific decrease in biogas yields from grape stalks (225 lN biogas kg $VS^{-1}$) to their high lignin content, which could not be hydrolyzed during the anaerobic digestion. Moreover, the inherent lignocellulosic nature of grape stalks, coupled with their significant mineral composition, offers a promising avenue for the conversion of cellulose into both C5 and C6 sugars. This conversion process holds the potential to yield essential platform chemicals [11]. Recent advancements in the bioconversion of lignocellulose into biofuels and value-added chemicals, as embraced within the Biorefinery Concept (effective utilization of lignocellulosic-based feedstock, such as wood, grass, and straw, for the production of a wide range of products), have garnered attention and extensive discussion. Sriariyanun et al. [12] delved into various technological considerations and economic facets pertaining to ongoing laboratory-scale research, emphasizing the need for knowledge transfer to facilitate industrial-scale production. This transition to a feasible process for the global lignocellulosic ethanol market is crucial. Furthermore, strategies to bolster the market influence of small and medium-sized enterprises (SMEs) have been explored. Examples include the creation of high-value products, co-production techniques, leveraging government support policies, and fostering cooperative networks among SME proprietors and local communities [13].

Aravindhan et al. [14] used grape stalks as a raw material to prepare activated carbon through an appropriate chemical synthesis route due to its high ash content. In addition, raisins contain antioxidant materials such as polyphenols, mainly flavonols (quercetin and kaempferol), that can be important in preventing cancer, AIDS, and coronary heart diseases [15]. Therefore, in the context of the increasing amounts of raisins produced and considering the resulting byproducts, the main aim of this study is to address the potential environmentally sustainable use of byproducts resulting from the processing of grapes to produce raisins. This study aims to develop processes to minimize the disposal of byproducts and maximize the added value.

## 2. Materials and Methods

### 2.1. Preparation of Byproduct Samples

During the process of raisin production from Thompson Seedless grapes, eleven different byproducts were collected from grape drying and plant processing, which was being carried out in the northern part of Egypt. A solar–biogas hybrid drying system was used as the fruit drying facility to dry different types of fruits, including grapes. Numerous manual and mechanical processes were carried out to obtain the required raisin quality ready for packaging; in each process, byproducts were generated in different quantities and varied characteristics. After drying, byproducts 1–4 were manually separated using four different sieves, and they contained bunches, grape stalks, rachises, large peduncles and pedicels. During washing and cleaning, laborers separated the small peduncles, pedicels and other materials using two processes, producing byproducts 5 and 6 that are characterized by small-size peduncles and pedicels.

Three different levels of air suction cleaning were used during the mechanical cleaning stage to separate raisin and non-raisin materials and thereby produce byproducts 7, 8, and 9, which consist of broken and small-size raisins and non-raisin materials such as small-size peduncles and pedicels. The remaining byproducts (10 and 11) were the rejected raisins that failed to fulfill the quality standards in the sorting and grading stage. Figure 1 presents the flowchart of byproducts collected at different stages of raisin production.

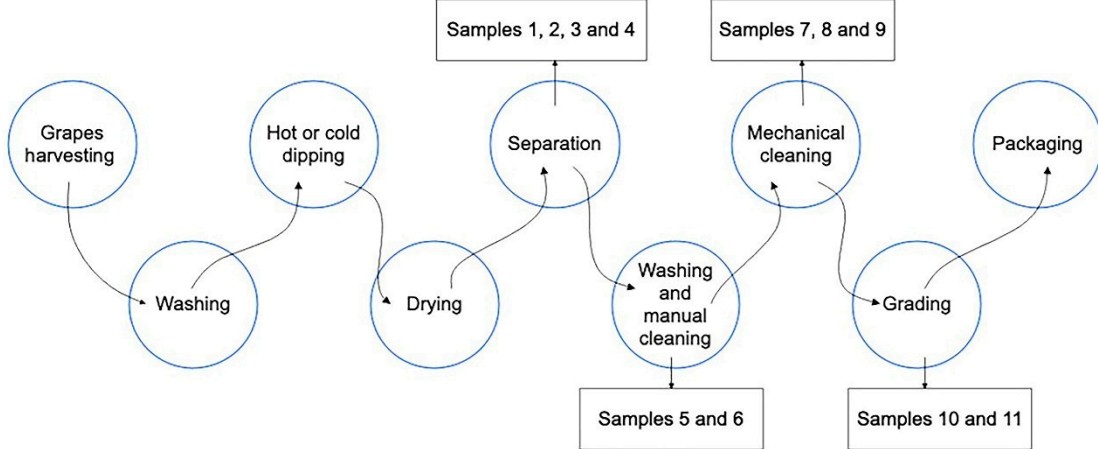

**Figure 1.** Process flowchart for raisin production stages, including samples of raisin byproducts generated at every stage.

*2.2. Ultimate Analysis and Heating Value*

The elemental analysis of carbon (C), hydrogen (H), sulfur (S), and nitrogen (N) contents in different samples was performed using the CNHOS element analyzer (X-ray fluorescence spectrometry (XRF), Malvern Panalytical Almelo, the Netherlands). The oxygen (O) percentage was estimated by the difference: O (%) = 100% − C (%) − H (%) − N (%) − S (%) − Ash (%). During the anaerobic digestion, the pH of the inoculums was measured by a portable pH Oakton EcoTestr meter (Cole-Parmer 625 East Bunker Ct Vernon Hills, Illinois, USA). The heating value was expressed as HHV (higher heating value) of the precise dry results. The byproducts' lower heating value (LHV) was calculated by subtracting the latent heat of water vaporization from the HHV. The calculations were determined according to the standard procedures for determining the higher heating value of solid and liquid fuels using the isoperibol or static-jacket Calorimeter (gross calorific value, DIN 51900-2, 2003) and the bomb calorimeter XRY-1A model (Huanghua Faithful Instrument, Chongqing, China). The calorimetric bomb was prepared using standard benzoic acid to calibrate the calorimeter. The byproduct samples were air-dried, milled, and sieved. A sample of 1 g of each byproduct was tested and the test was replicated 3 times to obtain dry samples using the elemental oxygen, nitrogen, and hydrogen contents of the byproducts [16]. The ash content was determined using the thermogravimetric method (EN ISO 18122) by the dry combustion of 1.5 g of the different waste samples by increasing the temperature at a rate of 20 °C min$^{-1}$ to 950 °C and holding for 60 min at this temperature [17].

*2.3. Chemical Composition*

The samples were treated and analyzed at the Regional Center for Food and Feed (RCFF), Agricultural Research Center (ARC), Ministry of Agriculture and Land Reclamation (MALR) in Egypt. The samples were oven-dried, milled, and sieved according to EN ISO 14780:2017. Samples with an average particle size of 0.5 were used in the tests. Total sugar was measured by HPLC using a 300 mm × 4 mm ID μ Bondapak/Carbohydrate column (water) with an RI detector. The eluent used was a mixture of acetonitrile:water (80:20) at a flow rate of 2 mL per minute and a temperature of 25 °C. Protein content was determined by the Bradford method [18] using distilled water as the blank and bovine serum albumin (BSA) as the standard solution for the standard curve by referring to the calibration curve. The amount of carbohydrate in each waste sample was determined using the phenol sulfuric acid method [19] using 2 g/L of glucose as a standard at various concentrations (0 to 0.25 g/L with every interval of 0.05 g/L of glucose) for the standard curve. The amount of carbohydrate in each waste sample was determined using the calibration curve. The moisture content and crude fiber content were determined by the

gravimetric method [20]. For potassium (K) determination, sulfuric acid was used to digest the prepared sample at 360 °C for 2 h. The total potassium concentration was determined by complete oxidation of the sample, followed by spectrometric analysis [21].

### 2.4. Lignocellulose Composition and Total Solids Concentrations

The standard methods were used to determine neutral detergent fiber (NDF), acid detergent fiber (ADF), and acid detergent lignin (ADL) [22]. The hemicellulose (HC) content was calculated as NDF − ADF, and celluloses (CE) as ADF − ADL [23]. Total solids (TS) and volatile solids (VS) were determined for byproduct samples after 24 h at 105 °C and after 4 h at 550 °C in the furnace [24]. The TS and VS of the inoculums were also identified in anaerobic digestion experiments.

### 2.5. Biogas Production

Biogas experiments were performed in pilot batches of the anaerobic digestion (AD) system under mesophilic conditions. The AD system was earlier installed in the plant's processing unit comprising a hybrid solar–biogas drying facility. The AD system consists of three digesters, each with a volume of 50 L and equipped with appropriate methods for collecting gases and residues. The system also includes feeding and smooth mixing devices to facilitate the fermentation of the class of byproducts used. The eleven types of byproducts were classified into three categories (A: byproducts from 1 to 4, B: byproducts from 5 to 9, and C: byproducts from 10 to 11) due to the absence of significant differences between byproducts of the same group. The digesters were filled with an inoculum mixture (animal manure-based digestion) and byproducts (substrates). A barrier consisting of 10 L of water, 200 mL of 1.00 N standard potassium dichromate solution, and 50 mL of 95% concentrated sulfuric acid was used in a container to receive the produced gas and avoid $CO_2$ diffusion [25]. The biogas produced was determined by noting the equivalent volume of liquid displaced from the gas collection container to the graduated empty container. The volume of overflown liquid was recorded daily for 40 days, and the recorded biogas production was expressed as the average volumetric production rate (L biogas/L substrate). The other aspects of biogas production, measurement, and estimation were considered [26]. Methane (CH4) production was measured using a portable gas analyzer GA500 (Geotech, Leamington Spa, Surrey, UK), adapted to measure CH4 accurately.

### 2.6. Statistical Analysis and Data Visualization

Data from the ultimate analysis, heating values experiments, and chemical composition were statistically analyzed using XLSTAT software (statistical add-in for Microsoft Excel) version 2019 (Addinsoft Inc., New York, NY, USA). Pairwise comparison using the Tukey (HSD) test was used to determine the significant differences among the results of byproducts analysis. The descriptive statistics including the mean and standard deviation of three replicates were calculated for each byproduct. The cumulative biogas production was presented using Excel's running total (cumulative sum). ANOVA and Duncan's test were used to evaluate significant differences between means at a significance level of 0.05. The data are presented as the mean ± SE of three replicates.

## 3. Results and Discussion

### 3.1. Ultimate Analysis and Heating Values

The ultimate analysis results for the raisin byproducts are listed in Table 1. According to ANOVA analysis, C, N, H, and O were significantly different among byproducts ($p < 0.05$); however, the difference was not significant for S ($p > 0.05$). The results revealed that the carbon (C) percentage ranged from 40.82 to 46.43%, meaning that fulvic acids constitute the carbon percentage of all byproducts. High carbon contents in byproduct samples from raisin production translated into high heating values [14]. The hydrogen (H) content ranged from 5.18 to 6.13% in the measured byproduct samples with slight differences, as shown in Table 1. Table 1 shows a significant reduction in nitrogen (N) and

sulfur (S) content, which varied from 1.41 to 2.68% for N and from 0.07 to 0.17% for S. Low S and N contents in alternative solid fuels are welcomed as positive results because it results in less release of sulfur and nitrogen oxides into the atmosphere, indicating that the burning of biomass evaluated in this study will not contaminate the environment, based on sulfur emission limits not exceeding 0.10%, as recommended by the Exhaust Gas Cleaning System Association (EGCSA) regulation to reduce a 0.50% sulfur cap [27], and consistent with the call from the United Nations Environment Programme to reduce the impacts of excess nitrogen on environmental quality and ecosystem services [28].

**Table 1.** Elemental analysis of the tested byproducts (dry mass).

| Byproduct | Parameter | | | | | |
|---|---|---|---|---|---|---|
| | C (%) ± SD | N (%) ± SD | H (%) ± SD | S (%) ± SD | O * (%) ± SD | Ash (%) ± SD |
| Byproduct 1 | 40.82 ± 0.03 g | 1.82 ± 0.09 f | 5.60 ± 0.04 c | 0.12 ± 0.03 a | 43.85 ± 0.03 b | 7.79 ± 0.12 a |
| Byproduct 2 | 45.62 ± 0.02 ab | 1.98 ± 0.12 ef | 5.18 ± 0.04 e | 0.17 ± 0.05 a | 41.25 ± 0.16 e | 5.80 ± 0.07 c |
| Byproduct 3 | 44.17 ± 0.01 cd | 2.08 ± 0.11 de | 5.27 ± 0.05 de | 0.12 ± 0.04 a | 42.51 ± 0.06 cd | 5.85 ± 0.07 c |
| Byproduct 4 | 45.95 ± 0.03 a | 2.48 ± 0.05 b | 5.38 ± 0.03 d | 0.10 ± 0.01 a | 39.31 ± 0.07 f | 6.78 ± 0.04 b |
| Byproduct 5 | 43.21 ± 0.01 e | 1.99 ± 0.04 ef | 5.89 ± 0.03 b | 0.10 ± 0.05 a | 43.70 ± 0.10 b | 5.11 ± 0.03 e |
| Byproduct 6 | 46.43 ± 0.01 a | 1.41 ± 0.06 g | 6.07 ± 0.05 a | 0.11 ± 0.03 a | 41.41 ± 0.07 e | 4.57 ± 0.06 g |
| Byproduct 7 | 44.93 ± 0.06 bc | 2.68 ± 0.03 a | 6.06 ± 0.05 a | 0.07 ± 0.03 a | 41.79 ± 0.09 de | 4.47 ± 0.04 g |
| Byproduct 8 | 43.65 ± 0.1 de | 1.99 ± 0.04 ef | 6.04 ± 0.06 a | 0.09 ± 0.03 a | 43.48 ± 0.10 b | 4.75 ± 0.05 f |
| Byproduct 9 | 43.16 ± 0.01 e | 2.36 ± 0.04 bc | 5.80 ± 0.02 b | 0.15 ± 0.03 a | 43.14 ± 0.06 bc | 5.40 ± 0.04 d |
| Byproduct 10 | 42.25 ± 0.03 f | 2.04 ± 0.04 e | 6.06 ± 0.02 a | 0.13 ± 0.05 a | 45.85 ± 0.14 a | 3.65 ± 0.06 i |
| Byproduct 11 | 43.67 ± 0.01 de | 2.28 ± 0.04 cd | 6.13 ± 0.03 a | 0.15 ± 0.04 a | 43.73 ± 0.02 b | 4.05 ± 0.05 h |

SD is standard deviation; a, b, c, d, e, f, g, h, and i refer to significant differences at $p < 0.05$ according to a pairwise comparison using Tukey (HSD) test. * Calculated values.

The ash content of byproducts 5–11 ranges from 3.65 to 5.11%, which is lower than the other values for the other byproducts in this study, as shown in Table 1. The energy content of raisin biomass is determined by its heating value (calorific value). The heating value is influenced by the biomass elemental composition, moisture, and ash content. Interestingly, the lower ash content was more marked in byproducts 9 and 10 than in the other byproducts. This result is consistent with [29], who found that the ash content of the lateral stems was higher than that of the fruit. This could be because rejected raisins are mainly composed of lignin, which is more resistant to hydrothermal carbonization conversion, whereas branches and grape stalks have higher hemicellulose and cellulose contents which produce a lot of ash during combustion. For the same reason, byproducts 2 and 3 have lower calorific values than the other byproducts [30,31].

Lower and higher heating values (LHV and HHV) of the tested byproducts are presented in Figure 2. For grape wastes, the HHVs of grape pomace, grape marc, and grape seeds were measured to be 20.34, 14.60–17.75, and 19.78–21.13 MJ kg$^{-1}$, respectively [32]. The low heating value ranged from 17.12 to 20.20 MJ kg$^{-1}$, as shown in Figure 2. These values are significantly higher than those reported in a previous study of grape residue biomass [33,34], which measured values of 11.82 and 14.39–16.66 MJ kg$^{-1}$ for vineyard pruning residue in Serbia and 7.34–7.96 MJ kg$^{-1}$ for vineyard pruning residue in Italy, respectively. The differences in heating value, whether HHV or LHV, are mainly due to different carbon content (main energy source) and different ash content (non-combustible material) [35].

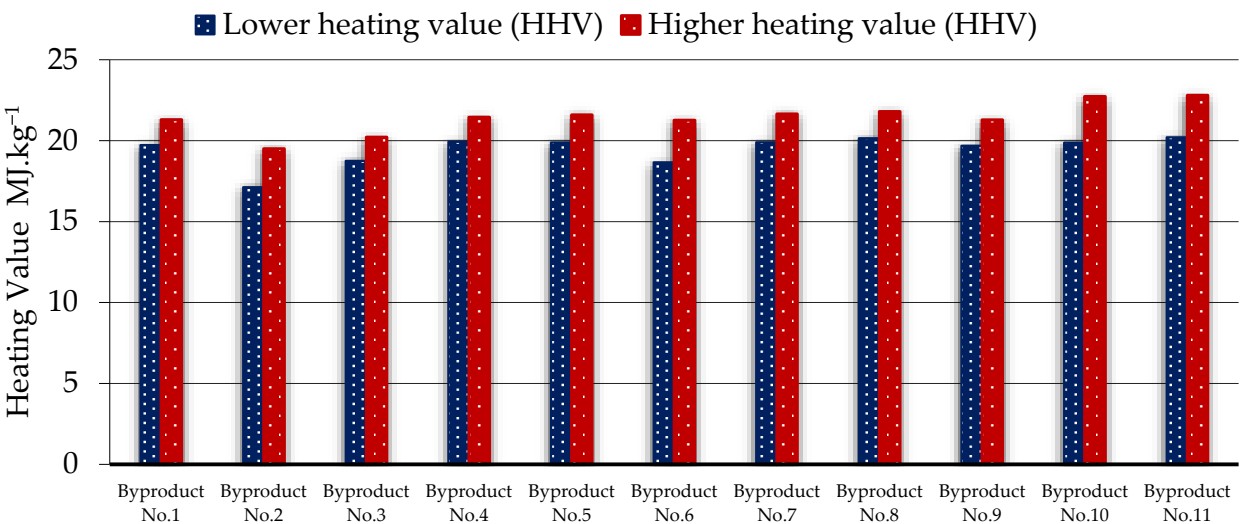

**Figure 2.** Lower heating value (LHV) and higher heating value (HHV) in MJ kg$^{-1}$ of biomass generated as byproducts at various stages of grape drying process.

### 3.2. Chemical Compositions

The chemical compositions of the byproduct samples are summarized in Table 2. It is known that the chemical compositions of raisin byproducts differ based on the grape variety, growing climates, and processing conditions. The moisture samples in this study fluctuated within a limited range of values from 4.41 to 9.24%, which is very low compared with values reported by Guerini et al. [30]. Byproducts 10 and 11 had the highest moisture content (9.15–9.24% d.b.), while byproduct 1 had the lowest moisture level (4.41% d.b.). The significantly higher moisture content of byproducts 7–11, as shown in Table 2, leads to lower energy emitted per kg, which results in lower calorific values. This is consistent with the lower carbon content as determined by the ultimate analysis, carbon content being a predictor of calorific value, as also indicated by Karacan and Olea [36].

**Table 2.** Chemical composition of byproduct samples at different stages of raisin production.

| Byproduct | Parameter | | | | | |
|---|---|---|---|---|---|---|
| | Moisture Content (%) ± SD | Total Sugar (%) ± SD | Total Carbohydrates (%) ± SD | Fiber (%) ± SD | Protein Content (%) ± SD | K (%) ± SD |
| Byproduct 1 | 4.41 ± 0.3 h | 26.86 ± 0.06 d | 46.55 ± 0.03 e | 4.20 ± 0.02 b | 9.43 ± 0.28 f | 1.29 ± 0.2 b |
| Byproduct 2 | 5.11 ± 0.05 g | 27.18 ± 0.04 d | 46.12 ± 0.05 f | 3.81 ± 0.02 c | 10.51 ± 0.03 d | 1.20 ± 0.02 cd |
| Byproduct 3 | 4.54 ± 0.04 h | 24.42 ± 0.10 f | 46.24 ± 0.07 f | 4.22 ± 0.07 b | 9.40 ± 0.04 f | 1.27 ± 0.01 bc |
| Byproduct 4 | 6.19 ± 0.03 e | 24.35 ± 0.16 f | 46.27 ± 0.06 f | 4.60 ± 0.25 a | 9.90 ± 0.06 e | 1.10 ± 0.07 e |
| Byproduct 5 | 5.65 ± 0.04 f | 21.83 ± 0.60 g | 47.27 ± 0.04 d | 3.85 ± 0.02 c | 12.21 ± 0.03 b | 1.017 ± 0.01 f |
| Byproduct 6 | 5.65 ± 0.03 f | 25.27 ± 0.05 ef | 48.51 ± 0.04 c | 3.39 ± 0.02 e | 10.26 ± 0.09 d | 1.02 ± 0.01 f |
| Byproduct 7 | 7.90 ± 0.02 c | 28.14 ± 0.03 d | 49.27 ± 0.04 b | 2.60 ± 0.02 ef | 12.41 ± 0.12 b | 1.07 ± 0.01 ef |
| Byproduct 8 | 7.27 ± 0.02 d | 26.60 ± 0.07 de | 47.19 ± 0.04 d | 2.85 ± 0.06 e | 12.16 ± 0.01 b | 1.38 ± 0.02 a |
| Byproduct 9 | 8.88 ± 0.03 b | 29.55 ± 0.22 c | 48.52 ± 0.08 c | 2.50 ± 0.05 f | 11.80 ± 0.02 c | 1.18 ± 0.01 d |
| Byproduct 10 | 9.15 ± 0.02 ab | 39.28 ± 0.04 b | 49.79 ± 0.13 a | 1.80 ± 0.04 g | 13.77 ± 0.05 a | 1.09 ± 0.02 e |
| Byproduct 11 | 9.24 ± 0.03 a | 41.18 ± 0.03 a | 49.80 ± 0.06 a | 2.03 ± 0.03 g | 13.93 ± 0.05 a | 1.08 ± 0.01 ef |

SD is standard deviation; a, b, c, d, e, f, g and h refer to significant differences at $p < 0.05$ according to pairwise comparison using Tukey (HSD) test.

Raisin waste is considered an important source of protein, and its byproducts contain 9.40%–13.93% protein, as indicated in Table 2. The total protein content of raisin waste may vary significantly depending on the type of byproducts, grape variety, location, and

fertilization conditions [37]. Byproducts 1, 3, and 4 had high protein content, up to 9.90%, which is similar to the values for byproducts 2, 6, and 9. Higher protein values were found in byproducts 5, 7, and 8. However, the greatest values were achieved in the byproducts 11 and 12, which are higher than for the raisin residues indicated in the previous studies. According to Llobera and Cañellas [38], stems have a low sugar content, which was also observed in this study. Byproducts 1 to 9 containing bunches, grape stalks, rachises, and different sizes of peduncles and pedicels had low sugar contents compared with byproducts 10 and 11 consisting of rejected raisins. Also, the main characteristics of the raisin byproducts were their high carbohydrate content (above 46%) and low fiber content, especially byproducts 10 and 11, as shown in Table 2.

*3.3. Lignocellulose Composition of Byproduct Samples*

The lignocellulose composition of byproduct samples is presented in Table 3. A lignin content of 26.35 ± 0.67% was reported by Kell [39] for grape pomace, which is slightly higher than the lignin content (22.23–23.55%) obtained in this study. Byproducts 5–9 had the lowest lignin content compared with other byproducts.

**Table 3.** Lignocellulose composition of byproduct samples at different stages of raisin production.

| Byproduct | Parameter | | | | |
|---|---|---|---|---|---|
| | NDF (% TS) ± SD | ADF (% TS) ± SD | ADL (% TS) ± SD | CE * (% TS) ± SD | HC * (% TS) ± SD |
| Byproduct 1 | 61.62 ± 0.02 a | 46.24 ± 0.07 abc | 23.24 ± 0.07 cd | 22.10 ± 0.01 c | 15.56 ± 0.08 ab |
| Byproduct 2 | 61.80 ± 0.08 a | 46.43 ± 0.08 ab | 23.37 ± 0.04 bc | 23.06 ± 0.04 bc | 15.19 ± 0.15 bc |
| Byproduct 3 | 61.30 ± 0.07 ab | 46.78 ± 0.17 a | 23.26 ± 0.08 cd | 23.52 ± 0.18 ab | 14.52 ± 0.34 de |
| Byproduct 4 | 61.57 ± 0.18 a | 46.52 ± 0.09 ab | 23.55 ± 0.01 a | 22.98 ± 0.09 c | 15.05 ± 0.24 bcd |
| Byproduct 5 | 61.09 ± 0.58 ab | 45.17 ± 0.09 d | 22.38 ± 0.04 g | 22.79 ± 0.12 c | 15.92 ± 0.51 a |
| Byproduct 6 | 60.70 ± 0.05 b | 46.00 ± 0.55 bc | 22.37 ± 0.07 g | 23.63 ± 0.48 a | 14.70 ± 0.11 cd |
| Byproduct 7 | 58.32 ± 0.06 d | 45.83 ± 0.08 c | 22.79 ± 0.05 f | 23.05 ± 0.07 bc | 12.48 ± 0.09 f |
| Byproduct 8 | 58.29 ± 0.11 d | 44.33 ± 0.03 e | 22.96 ± 0.07 e | 21.37 ± 0.04 e | 13.96 ± 0.09 e |
| Byproduct 9 | 59.39 ± 0.06 c | 44.50 ± 0.20 e | 22.23 ± 0.07 g | 22.28 ± 0.13 d | 14.89 ± 0.26 cd |
| Byproduct 10 | 55.39 ± 0.92 e | 43.68 ± 0.03 f | 23.20 ± 0.05 d | 20.48 ± 0.02 f | 11.71 ± 0.08 g |
| Byproduct 11 | 55.24 ± 0.07 e | 43.66 ± 0.10 f | 23.49 ± 0.07 ab | 20.17 ± 0.12 f | 11.58 ± 0.03 g |

SD is standard deviation; a, b, c, d, e, f, and g refer to significant differences at $p < 0.05$ according to pairwise comparison using Tukey (HSD) test. * Calculated values.

The provided statistical analysis in Table 3 presents data on various parameters measured across different byproducts. Each row represents a specific byproduct, while the columns show the average values (with standard deviations) of the parameters studied. Grape stalks are composed of lignocellulosic compounds, such as cellulose (CE) and hemicelluloses (HC) [40] and a relatively high content of lignin (between 22% and 47%) [41]. In this study, the lignin content of byproduct 2 (containing grape stalks) was slightly higher than the CE content, which was 23.06%, and significantly higher than the HC value (15.19%), as shown in Table 3. The concentration of these compounds in grape stalks differs according to geographic origin, climate, time of harvest, and grape varieties, which is why the lignocellulose composition of byproduct samples here may differ from the values determined by different researchers. There were no significant differences in neutral detergent fiber (NDF) contents between products 1 to 6, while there were slight differences between products 7 to 11.

Acid detergent fiber (ADF) contents were numerically higher for byproducts 1–7, in agreement with previous lignocellulose composition investigations, which reported that the raisin wastes containing stalk (mainly pedicels and rachises) had higher contents of ADF and NDF. However, relatively lower values were found for byproducts 8–11 (containing (small-size peduncles and pedicels and rejected raisins) with average values of 44.33, 44.50, and 43.66, respectively (Table 3). The lower levels of ADF in byproducts 8–11 could be because they contain non-structural carbohydrates that are mostly found in berries and mainly contain simple sugar, pectin, and starch.

### 3.4. Total Solids Concentrations

The total solids concentrations of byproduct samples as a substrate or inoculum are shown in Table 4. The maximum TS is recorded in byproduct 10 (35.20%). The VS in all byproducts examined in this study varies from 91.54% (byproduct 11) to 93.82% (byproduct 1). The characteristics of the TS and VS inoculums studied in this paper are as follows: TS (7.29–7.66%), and VS (79.23–81.37%). The concentrations of TS and VS inoculums reported by different researchers are 9.30% TS and 80.3% VS reported by Chua et al. [42] and 16% TS reported by Deressa et al. [43].

**Table 4.** Total solids concentrations of byproduct samples and inoculums.

| Byproduct | Parameter | | | | |
|---|---|---|---|---|---|
| | TS (%) Byproduct | VS (% TS) Byproduct | TS (%) Inoculum | VS (% TS) Inoculum | pH Inoculum |
| Byproduct 1 | 31.70 ± 0.04 d | 93.82 ± 0.06 a | 7.66 ± 0.02 f | 81.37 ± 0.06 a | 7.64 ± 0.01 c |
| Byproduct 2 | 31.30 ± 0.04 d | 93.30 ± 0.07 bc | 7.58 ± 0.01 cd | 81.23 ± 0.05 a | 7.58 ± 0.01 d |
| Byproduct 3 | 31.22 ± 0.06 d | 93.18 ± 0.07 c | 7.62 ± 0.01 bcd | 80.77 ± 0.27 b | 7.72 ± 0.01 b |
| Byproduct 4 | 31.37 ± 0.04 d | 93.49 ± 0.06 b | 7.72 ± 0.01 a | 81.22 ± 0.06 a | 7.35 ± 0.01 e |
| Byproduct 5 | 34.53 ± 0.05 c | 92.78 ± 0.11 d | 7.65 ± 0.03 abc | 79.37 ± 0.04 c | 7.77 ± 0.01 a |
| Byproduct 6 | 34.61 ± 0.07 c | 92.84 ± 0.09 d | 7.56 ± 0.04 d | 79.23 ± 0.07 c | 7.12 ± 0.02 f |
| Byproduct 7 | 34.88 ± 0.04 bc | 92.67 ± 0.04 d | 7.43 ± 0.01 e | 79.32 ± 0.07 b | 7.36 ± 0.01 e |
| Byproduct 8 | 34.84 ± 0.54 bc | 92.08 ± 0.18 e | 7.40 ± 0.03 e | 80.63 ± 0.02 d | 6.81 ± 0.01 g |
| Byproduct 9 | 34.61 ± 0.06 c | 92.25 ± 0.08 e | 7.31 ± 0.03 f | 78.69 ± 0.02 d | 6.48 ± 0.02 i |
| Byproduct 10 | 35.20 ± 0.04 ab | 91.29 ± 0.05 g | 7.32 ± 0.03 f | 78.59 ± 0.05 d | 6.90 ± 0.01 g |
| Byproduct 11 | 35.46 ± 0.05 a | 91.54 ± 0.07 f | 7.29 ± 0.06 f | 79.36 ± 0.02 c | 6.74 ± 0.01 h |

SD is standard deviation; a, b, c, d, e, f, g, h and i refer to significant differences at $p < 0.05$ according to pairwise comparison using Tukey (HSD) test.

The amount of volatile solids (VS) indicates the suitability of waste as a feedstock for pyrolysis conversion to a liquid product, where a high volatile content is desirable. All byproducts (1–11) have high contents of volatile solids (VS) (Table 4). As a result, these fuels can easily ignite and subsequently oxidize. High volatile matter also contributes to improved burning during combustion [44].

The pH values of the raisin byproducts were between 6.74 and 7.64, as shown in Table 4. This leads to more stability by hampering the development of microorganisms, especially fungi which prefer an acidic pH (4.5–5.0). However, it is a suitable environment for bacterial growth as bacteria prefer a near-neutral pH (6.5–7.0) [37].

Optimal conditions for anaerobic digesters are run within in pH range of 7.0–8.5; outside this range, imbalances can occur, and these values were achieved in byproducts 1 through 7. In addition, methane production is reported to cease once the pH drops below 6.2 [45]. Therefore, the stability of the normally volatile system is improved, resulting in improved biogas and methane production.

### 3.5. Biogas Production and Methane Percentages

The biogas potential of the tested byproduct groups collected from raisin production stages was assessed. The average values of daily biogas production of each byproduct group are presented in Figure 3A. Daily biogas production started on the first day of digestion with peak production in the first 6 days. The biogas production decreased until it reached almost zero after 40 days, showing a similar trend for the three tested byproduct groups A, B, and C. The highest daily biogas production rate reached 1.04, 0.80, and 0.74 NL L$^{-1}$ of substrate day$^{-1}$ on the third digestion day for byproduct groups C, B, and A, respectively. The daily biogas production started to decrease during the period of 7–30 days, with production fluctuating from 0.49 to 0.11 NL L$^{-1}$ of substrate day$^{-1}$ for byproduct group C, and from 0.44 to 0.12 NL L$^{-1}$ of substrate day$^{-1}$ for byproduct group B, and from 0.40 to 0.10 NL L$^{-1}$ of substrate day$^{-1}$ for byproduct group C. Generally,

there was a sharp increase in biogas production in the first three days of the AD process, followed by a gradual decline until there was no production. The availability of substrates at the early stages of AD could be a factor, and as AD progressed, the substrates on which the microorganisms could feed were depleted [46]. The patterns in average daily biogas production for each byproduct group were very similar to those reported by Olugbemide and Likozar [47].

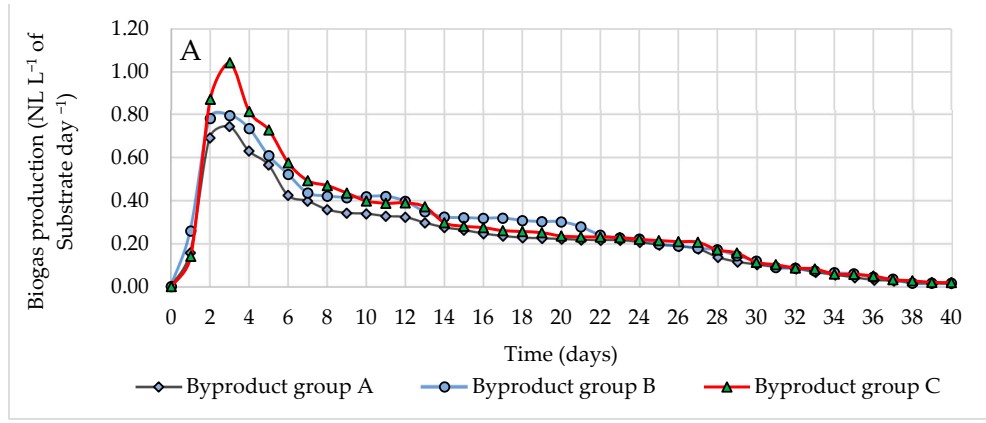

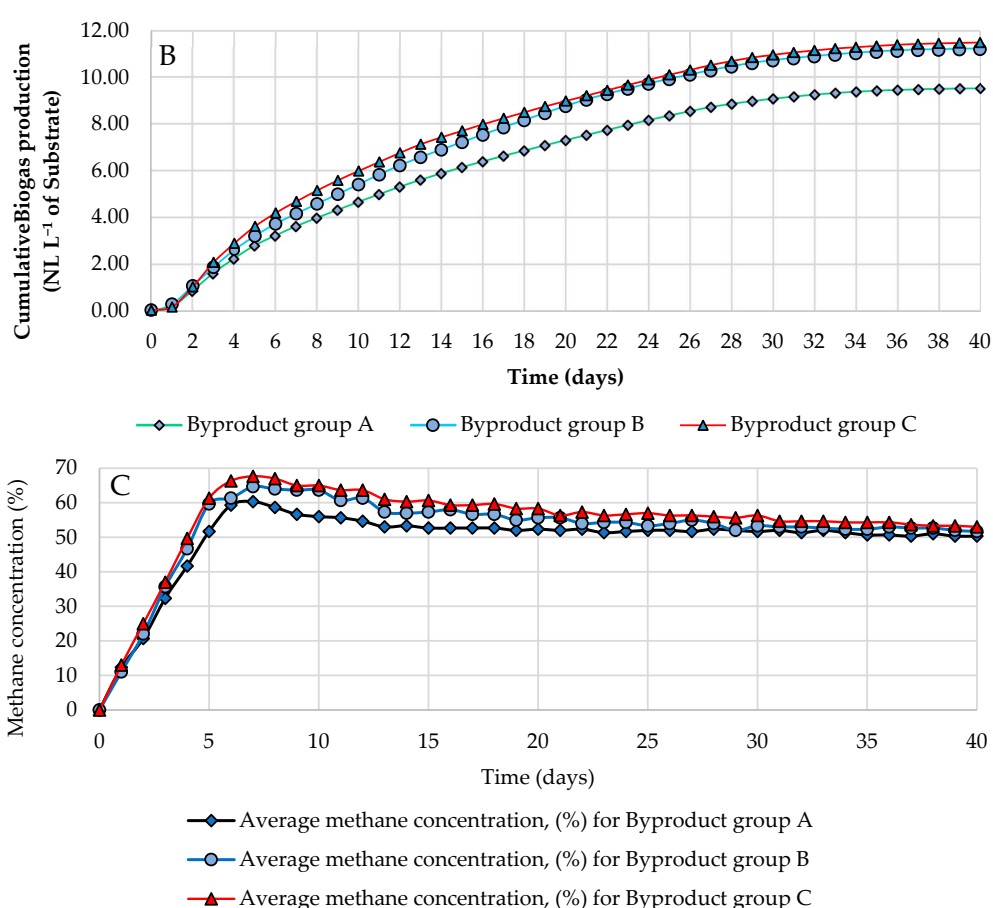

**Figure 3.** (**A**) Average daily biogas production for each byproduct group; (**B**) Average accumulated biogas production for 40 days for each byproduct group; and (**C**) methane content in the biogas for each byproduct group.

The average accumulated biogas production for 40 days for each byproduct group is presented in Figure 3B. The highest accumulated biogas volume after 40 days was 11.50 NL L$^{-1}$ of substrate for byproduct group C followed by 11.20 NL L$^{-1}$ of substrate for byproduct group B and 9.51 NL L$^{-1}$ of substrate for byproduct group A.

The composition of the organic matter in terms of the cellulose concentration and lignin significantly impacts the biogas produced by the tested byproducts. The byproduct group with high carbohydrates (byproduct group C: 49.79 ± 0.13 for byproduct 10, and 49.80 ± 0.06 for byproduct 11) is characterized by its high degradability and rapid transformation, resulting in higher biogas yield. The methane content of biogas is a crucial parameter, especially regarding its applicability. Biogas predominantly consists of methane (50–70%) and carbon dioxide (30–40%). The methane content of the biogas for each byproduct group is presented in Figure 3C. The percentage of methane in biogas gradually increased to the highest value during the first week, then stabilized between 50 and 60%. A double curvature shape was observed for methane production curves that can be linked to generally slow degradation of the main components at the beginning followed by the maximum increase during days 6 to 10. Furthermore, the shape of the curves may indicate that the digestion process occurred in two steps as a portion of the organic materials needed a longer time for hydrolysis and degradation into methane [48,49]. The highest methane concentrations were 68, 65, and 60% for byproduct groups C, B, and A, respectively. The slight differences in methane content between the three groups A, B, and C result from the different compositions of the fermented mixture.

## 4. Conclusions

This conclusive investigation underscores the promising potential of utilizing biomass derived from raisin production as a dependable source of solid fuel. Particularly noteworthy are the lower ash content and significantly elevated calorific values observed for specific byproducts, signifying their appropriateness for energy generation. Given the diminished moisture content in the raisin byproduct samples, their application as solid biofuels can be undertaken without necessitating moisture reduction or instigating elevated emissions of pollutants. The examination of the volatile solids content in the raisin processing byproducts indicates their prospective suitability for transformation via pyrolysis into a liquid product distinguished by a notable volatile composition. The onset of daily biogas production was observed immediately upon starting the digestion, demonstrating uniformity across the tested byproducts. These byproducts exhibit high methane concentrations with slight variations in methane content attributed to the heterogeneous composition of the fermented mixture. Therefore, this study recommends the biomass obtained from various stages of the grape drying process as a prime opportunity for biogas production.

**Author Contributions:** Conceptual idea, M.O., R.H. and R.K.; Methodology design, M.O. and R.H.; Data collection, R.H. and R.K.; Data validation, M.O. and R.H.; Data analysis and interpretation, M.O., R.H. and R.K.; Original draft preparation, M.O. and R.H.; Review and editing, M.O. and R.K.; Supervision and administration, M.O. and R.H. All authors have read and agreed to the published version of the manuscript.

**Funding:** This research received no external funding.

**Data Availability Statement:** Data will be made available upon request.

**Conflicts of Interest:** The authors declare no conflict of interest.

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
