# Peer review of "Assessment of Raisins Byproducts for Environmentally Sustainable Use and Value Addition"

_agriengineering, doi:10.3390/agriengineering5030091_

Round 1

Reviewer 1 Report

-52-53, authors should briefly explain about lignocellulose biorefinery and examples of products, please see references at http://jase.tku.edu.tw/articles/jase-202402-27-2-0001, http://ojs.kmutnb.ac.th/index.php/ijst/article/view/2744/2082,  

-172-175, low S and N contents and authors conclude that it will not contaminate environment. Authors should add some standard value to confirm this conclusion

-213-214, please indicate the criteria and reference to be used to justify which material is protein source 

- table 3, authors should add more discussion about the effect of this process on the lignocellulose composition. Currently authors just reported the values of analysis but no other discusssion

- 243-252, please explain the mechanism of this observed results

- 286-288, please add discussion to explain about this reduction

- fig 4, please discuss why the biogas production daily started to decline at 3rd day, while the methane concentration still stable after day 6th

-this manuscript should be revised by adding discussion and explain what's the difference to other published works. Authors should also collect and benchmark with other published papers about biogas production from lignocellulose biomass

minor checking is suggested

Reviewer 2 Report

The manuscript deals with an increasingly viable option for the transition towards a low-carbon bioeconomy. The results obtained from by-products from the raisin production process are attractive from an environmental and sustainable economic point of view. That is why I am in favor of publishing the Journal AgriEngineering manuscript after minor suggestions.

In the introduction I suggest adding a graph or table of the production of raisins and tailings in recent years to reinforce the objective of the research.

Make comments and comparisons of the results achieved in the research with others reported in the literature, either with the same residual biomass or with different ones. It can be tabulated so that readers can better observe the results obtained in comparison with others reported in other works. This made the work more attractive to readers and researchers.

Reviewer 3 Report

The manuscript “Assessment of Raisins Byproducts for Environmentally Sustainable Use and Value Addition” describes interesting research results.

Highlights and strengths of the manuscript are:

A thorough characterization of 11 raisins production by-products is described. The results may further increase interest in this waste and help develop new strategies for its use as solid fuel.

The experimental work has been carefully carried out, and analytical determinations were performed in triplicate.  The reference section is correct and includes relevant bibliography. 

Specific comments and suggested revisions:

-Line 158 should read "...standard deviation of three replicates".... 

-The claim in Line 174 regarding potential for contamination of the environment should be substantiated

-Line 320 should read "...by products 10 and 11".... 

-Figure 2 is difficult to interpret because all the pictures are similar. Perhaps the photos can be improved using higher definition?

This manuscript is clearly written. A minor revision could help improve the manuscript. 

Reviewer 4 Report

 This manuscript written by Okasha et al. investigated the potential and sustainable use of eleven distinct byproducts generated along with the processing of grapes to produce raisins.

Most of the parts of this manuscript are written well; however, some errors in grammar, phrases, and sentence structures were noticed throughout the manuscript.

Though, the information mentioned in the manuscript is sufficient. Some comments/ suggestions for the manuscript improvement are discussed below:

My vote is for acceptance after minor revision.

Introduction: this needs to be updated with new citations. The authors should work on the introduction section based on their topic.

Figure 2: need to increase the resolutions of the figure.

Conclusions: Is a very long

 The authors must concisely point out their conclusion and scientific contribution to the study.

References: 

style is not based on the format of the journal in some of them

Also, need to be updated.

minor editing
